# Cancer Stem Cell for Tumor Therapy

**DOI:** 10.3390/cancers13194814

**Published:** 2021-09-26

**Authors:** Binjie Huang, Xin Yan, Yumin Li

**Affiliations:** 1Department of General Surgery, Second Hospital of Lanzhou University, Lanzhou 730030, China; huangbj20@lzu.edu.cn (B.H.); yanx20@lzu.edu.cn (X.Y.); 2Key Laboratory of the Digestive System Tumors of Gansu Province, Second Hospital of Lanzhou University, Lanzhou 730030, China

**Keywords:** CSCs, mitotic division pattern, metabolic phenotype, therapeutic resistance, targeted strategy, tumor therapy

## Abstract

**Simple Summary:**

Although many methods have been applied in clinical treatment for tumors, they still always show a poor prognosis. Molecule targeted therapy has revolutionized tumor therapy, and a proper target must be found urgently. With a crucial role in tumor development, metastasis and recurrence, cancer stem cells have been found to be a feasible and potential target for tumor therapy. We list the unique biological characteristics of cancer stem cells and summarize the recent strategies to target cancer stem cells for tumor therapy, through which we hope to provide a comprehensive understanding of cancer stem cells and find a better combinational strategy to target cancer stem cells for tumor therapy.

**Abstract:**

Tumors pose a significant threat to human health. Although many methods, such as operations, chemotherapy and radiotherapy, have been proposed to eliminate tumor cells, the results are unsatisfactory. Targeting therapy has shown potential due to its specificity and efficiency. Meanwhile, it has been revealed that cancer stem cells (CSCs) play a crucial role in the genesis, development, metastasis and recurrence of tumors. Thus, it is feasible to inhibit tumors and improve prognosis via targeting CSCs. In this review, we provide a comprehensive understanding of the biological characteristics of CSCs, including mitotic pattern, metabolic phenotype, therapeutic resistance and related mechanisms. Finally, we summarize CSCs targeted strategies, including targeting CSCs surface markers, targeting CSCs related signal pathways, targeting CSC niches, targeting CSC metabolic pathways, inducing differentiation therapy and immunotherapy (tumor vaccine, CAR-T, oncolytic virus, targeting CSCs–immune cell crosstalk and immunity checkpoint inhibitor). We highlight the potential of immunity therapy and its combinational anti-CSC therapies, which are composed of different drugs working in different mechanisms.

## 1. Introduction

Tumors pose a significant threat to human life. The battle against tumors has always been fierce across human society, and many scientists are working on revealing the mechanisms of genesis, progression, recurrence and metastasis of the tumor. Many methods have been proposed to tackle tumors, such as operations, chemotherapy and radiotherapy; however, the consequences are not satisfactory. A previous study showed that cancer ranked 2rd among all of the leading causes of death in the US in 2018 [1]; it even ranked the first in some regions in the US [2]. In recent years, as a kind of revolutionized method, molecular targeted therapies showed many successful aspects in many different cancer types [3]. The key point of molecular targeted therapy is to find a perfect therapeutic target, which can be DNA, RNA, protein or some special molecules located on the tumor cell, and it should be specific, effective and easy to work for the targeted drugs. According to the latest reports, the strategies targeting cancer stem cells (CSCs) could be a promising therapeutic strategy [4].

CSCs, also named tumor-initiating cells (TICs), are a subgroup of tumor cells, although the percentage of CSCs is less than approximately 2% in all tumors [5]; many studies have proven that CSCs were closely related to the genesis, development, metastasis, recurrence and therapeutic resistance of tumor, which could be the reasons for tumor treatment failure [6,7,8]. CSCs harbor similar properties to stem cells, self-renewal and multipotential differentiation. In addition to self-renewal and multi-directional differentiation properties, CSCs also have a unique mitotic division pattern, unique metabolic phenotype and stubborn resistance to antitumor therapies.

## 2. Biological Characteristics

### 2.1. Mitotic Division Pattern of CSCs

In general, tumor cells showed proliferation, invasion and metastasis characteristics via rapidly symmetric mitotic division; however, different mitotic division modes were observed in CSCs. To adapt to different proliferative conditions and therapeutic stress, researchers found that CSCs could exhibit plasticity by changing their mitotic division pattern. In accordance with previous scientists, stem cells benefited from the facultative phenomenon of symmetric and asymmetric divisions, which enabled them to perpetuate themselves and generate differentiated progeny [9]. It was found that the self-renewal of stem cells could be determined by asymmetric mitosis, which was completed under the regulation of the extrinsic stem cell niche signal to orient the mitotic spindle perpendicularly to the niche surface and the building of intrinsic polarity axis to localize cell fate determinants asymmetrically in mitosis [10]. Some CSCs showed an asymmetric division pattern during their mitosis [11,12]; in this mitotic division pattern, CSCs could acquire cellular asymmetry during the cell division interphase to reorient the polarity axis of the spindle and generate asymmetric division via the reorientation of the distribution of the Par3/Par6/atypical protein kinase C complex and the microtubule-associated nuclear mitotic apparatus protein (NuMA)/LGN/Gαi complex [13]. Subsequently, an asymmetric spindle morphology will form during the division phase in which the apical half spindle is much larger in size than that of the basal half spindle; a progenitor CSC will divide into two differently sized cells; CSC, which harbors the capacity of self-renewal and differentiation potential, and a differential tumor cell, which enters the tumor cell cycle (Figure 1). Different mitotic divisions occur in different tumor types, it was revealed that the division pattern of glioma stem cells was symmetric division, not asymmetric division [14]. In symmetric mitosis, CSCs could be divided into two CSCs or two differentiated tumor cells. Interestingly, researchers found that symmetrical and asymmetrical division patterns could coexist in the oral CSCs [15]. The mitotic division pattern plays a crucial role in tumor development; Sandra G-L et al. suggested asymmetric division as a tumor suppressor and that the loss of asymmetric division phenotypes could contribute to tumorigenesis [9]. Several researchers have revealed that asymmetric division is not the main division pattern of CSCs and even contributed to the evolutionary disadvantage of CSCs during tumor development; however, the asymmetric division also helped CSCs to survive under different therapeutic stress [16]. In summary, the mitotic pattern of CSCs and their related contribution to tumorigenesis remain unclear and controversial; the different cancer types, tumor stages, CSC niche circumstances and different therapeutic stresses decide the different mitotic patterns of CSCs.

### 2.2. Metabolic Pattern of CSCs

As we know, the energy metabolism of the normal cell occurs in the mitochondrial matrix, and the main productive pathway is always the tricarboxylic acid cycle (TAC) coupled to oxidative phosphorylation (OXPHOS) of the carbon source, including glucose, fatty acid and sometimes protein, glycolysis is another metabolic pattern with properties of faster efficient energy production and lower efficient use of glucose. It was revealed in the Warburg effect, the tumor cell could change its energy metabolic pattern from OXPHOS to aerobic glycolysis to meet its higher metabolic needs due to its faster proliferation property [17]. CSCs are a subgroup of tumor cells, the energy metabolic pattern should be glycolysis according to the Warburg effect, but controversial views have appeared in different studies. It has been indicated in many studies that the metabolic pathway of CSCs should be aerobic glycolysis, which could be initiated by pyruvate dehydrogenase kinase1 (PDK1) to enhance stemness in mouse model; glycolysis was also found to contribute to breast cancer stem cells (BCSCs) and as a potential therapeutic strategy for breast cancer [18]. What’s more, glycolysis was observed to promote an increased stemness phenotype and tumorigenicity in basal-like breast cancer [19]; this metabolic pattern was also found in some other solid tumors. Meanwhile, some researchers have found that OXPHOS could act as a metabolic phenotype in CSCs; for instance, it was indicated that OXPHOS was highly activated in acute myeloid leukemia stem cells [20], it has also been suggested that it could be a strategy to target OXPHOS of BCSCs to reverse the multi-drug resistance (MDR) [21]; the OXPHOS metabolic pattern was also observed in glioblastoma cancer stem cells [22], lung CSCs [23] and some other cancer types. It is interesting that different metabolic patterns can coexist in the same tumor at the same time [24]. Many studies have indicated that the energy metabolic pattern is closely related to regulating the stemness of CSCs. CSCs show dynamic metabolic heterogeneity in different species, tumors, tumor microenvironments and CSC niches [25], which leads to therapeutic limitations in targeting the metabolic pathway of CSCs.

### 2.3. Therapeutic Resistance of CSCs

The tumor dormancy phenomenon has been found in many cancer types, and it has also been revealed that tumor dormancy contributes to therapy resistance, metastasis and immune system evasion; certain CSCs, called dormancy-competent CSCs (DCCs), share some characteristics with dormant tumor cells [26]. CSCs show resistance to conventional therapeutic drugs [27]; common chemotherapy drugs inhibit tumors by inducing DNA damage and inhibiting mitosis [28]; CSCs are not sensitive to chemotherapeutic drugs as they always stay dormant(G0 stage) [5], during which, the DNA damage-repair mechanism of CSCs will be initiated and mediates CSCs escaping from apoptosis [29].

Different concentrations of reactive oxygen species (ROS) play different roles in cell survival, of which, the low concentration of ROS can promote cell adaptive proliferation, the high one will cause permanent damage to DNA, RNA and the other bio-macromolecules in the cell; CSCs can survive in chemotherapy due to their low ROS level caused by the increased ROS scavenger level in CSCs [30].

ATP-binding cassette efflux transporters (ABC transporters) were membrane proteins which were firstly found in bacteria. ABC transporters were also found as multidrug-resistant (MDR) proteins because they could help cancer cells to avoid being eliminated by pumping chemotherapeutic drugs out [31,32] (Figure 2). An increased expression of ABC transporters (ABCB1, ABCC2 and ABCG2) was found in CSCs, which contributed to the MDR of CSCs. ABC transporters have been indicated as an ideal target in reversing the MDR of CSCs in many studies [33].

The tumor microenvironment (TME) harbors the capacity to maintain the stemness of CSCs; TME acts as a nest for CSCs, where CSCs can maintain their chemo/radiotherapy resistance via hypoxia, lower vascularization and different nutritional metabolic patterns [34]. Many methods have been proposed to reduce the resistance caused by CSCs. Shen et al. proposed a nanotherapeutic strategy, which was composed of all-trans retinoic acid, differentiation-inducing agent, and chemotherapeutics (camptothecin); it could reduce stemness-related resistance and prevent the mouse models from tumor relapse and metastasis after surgical operation [35]. Toni Nunes et al. proposed a material named gold nanoparticles, which can reverse resistance by functionalizing with antibodies to target CSCs [36].

Many other therapeutic resistance-related mechanisms have been proposed in detail in previous reviews, including increased autophagic activity, decreased ferroptosis, immune escape and favorable tumor vasculogenic mimicry in CSCs; all of these lead to the MDR and radio-resistance (RR) of CSCs in many tumor types [37].

## 3. Targeted Strategies of CSCs

### 3.1. Targeting Surface Molecules of CSCs

CSCs were firstly found in hematological malignancy, which were marked by CD34^+^/CD48^−^ [38]; after that, it was found that it was feasible to identify and isolate CSCs from tumor tissues via their surface markers. Subsequently, different CSC surface markers were found in different tumor types (Figure 3), and most of them were found as membrane proteins, which made these membrane proteins ideal antigens to be targeted by several specific antibodies. There is no doubt that the monoclonal antibody is the best choice due to its higher specificity and lower toxicity. The monoclonal antibody has proven its success in tumor therapy; the classic cases such as trastuzumab for HER2^+^ breast cancer and cetuximab for colorectal cancer are well known to us. According to the current knowledge, the monoclonal antibody could kill target cells via several different mechanisms, including antibody-dependent cytotoxicity and complement-dependent cytotoxicity, inhibiting cell signal transduction and inducing cell apoptosis directly. In this regard, researchers have focused on designing different monoclonal antibodies to target CSCs.

Adhesion molecules CD44, CD133, and CD90 were found as CSCs markers in many cancer types. Anti-CD44 was thought to be a potential targeting strategy in the treatment of CSCs [39]. Several monoclonal antibodies, such as H90 targeting CD44, 7G3 targeting CD123 and B6H12 targeting CD47, were proved effective in inhibiting leukemia stem cells [40]. RO5429083, a monoclonal antibody targeting CD44, has been evaluated in clinical trials in malignant solid tumors and acute myelogenous leukemia by Roche. The increased tumor-initiating potential of CD90 was observed in athymic nude mice, and the anti-CD90 monoclonal antibody showed its potential to inhibit the stemness of CSCs in malignant insulinoma [41]. There are also some other CSC markers, such as CD54, EpCAM and ALDH, which have been used to target CSCs; many clinical trials have been conducted in different areas, some of which are shown in Table 1.

The present problem is that the percentage of CSCs in cancer tissues is very small; meanwhile, the surface markers expressed on CSCs can also be expressed on normal cells (Table 2), meaning that the drugs targeting CSCs surface markers always show drug toxicity to healthy cells, which makes it difficult to target CSCs accurately. Thus, it is urgently needed to find more specific targeting markers to target CSCs in future research.

### 3.2. Targeting Signal Pathways of CSCs

In recent years, increased studies have proven that properties of CSCs could be regulated by different signal pathways, including Wnt, Notch, NF-kB, Hh, JAK-STAT, PI3K-AKT-mTOR, TGF-β and PPAR; most of these signal pathways are closely related to cell fate, and the abnormal activation of these signal pathways always contributes to tumor cell proliferation, differentiation, metastasis, recurrence and MDR via the regulation of CSCs in different tumor types. It was revealed that the Wnt signal pathway contributes to the stemness of CSCs, and some researchers have proposed that the stemness of CRC CSCs can be defined by high Wnt activity [81]. Paloma Ordóñez-Morán et al. found that the expression of HOXA5 could inhibit the activation of the Wnt pathway, induce the loss of CSC phenotypes, and prevent tumor development and metastasis in colon cancer [82]. The novel activation of Wnt and Notch signal pathway was observed in renal CSCs, and it was found that it could block the self-renewal and proliferation of renal CSCs by inhibiting such signal pathways [83]. The Hh signal pathway also contributes to the stemness of CSCs in several aspects; Mingli Zhou et al. indicated that the activated Hh signal pathway could enhance the expression of CSCs TFs (SOX2, OCT4) and they suggested that the stemness of CSCs could be gained by activating the Hh signal pathway [84]. Lipid desaturation was proved to be a metabolic marker for ovarian cancer stem cells, and its expression level could be directly regulated by the NF-kB signal pathway [85]. JAK2-STAT3 was overexpressed in the CRC stem cell, and the targeting gene cyclin D2 (CCND2) was increasingly transcribed to maintain the properties of CRC CSCs, such as proliferation and radio-resistance [86]. EMT, CSCs phenotypes and PI3K-AKT-mTOR signal pathway proteins were increasingly expressed in prostate cancer radio-resistant cell lines, and it was demonstrated that CaP radio-resistance was closely related to EMT and enhanced CSC phenotypes, which were mediated by the activation of the PI3K-AKT-mTOR signal pathway. Researchers found that the excessive activation of the TGF-βsignal pathway could enhance the stemness of CSCs in triple-negative breast cancer (TNBC); they suggested that the combination of classical chemotherapy agents and TGF-βinhibitors could improve the prognosis of TNBC [87]. Some non-steroidal anti-inflammatory drugs (NSAIDs), such as aspirin and indomethacin, could decrease the stemness of CRC CSCs by activating PPAP-γ [88]. Several other signal pathways have been less frequently reported in connection with CSCs; COX2 was found to be related to contributing to CSC phenotypes, such as apoptosis resistance, cell proliferation and MDR, via its metabolite prostaglandin E2, so researchers proposed that the COX2 signal pathway would be a potential target for anti-CSCs [89]. Obviously, the crucial role of CSCs in tumor development was regulated by many different signal pathways; thus, it is a logical strategy to eliminate CSCs and improve prognosis via inhibiting the signal pathways of CSCs [90].

#### 3.2.1. Notch Signal Pathway Inhibitor

##### γ-Secretase Inhibitors (GSIs)

As we know, the core step of the Notch signal pathway is the release of Notch intra-cellular domain(NICD). GSIs can prevent the release of NICD from the proteolytic cleavage of Notch receptors, GSIs have been proven to develop successfully in many different tumors and have also been suggested to be a targeting strategy in triple-negative breast cancer (TNBC) [91]; researchers have revealed that therapeutic sensitivity to GSIs could be influenced by NOTCH-1 mutations in TNBC [92]. A similar result was also observed in CRC [93]. RO4929097 is a kind of GSI; a single-agent strategy of RO4929097 showed its limitations in advanced tumors in many clinical trials, including metastatic melanoma [94], metastatic CRC [95] and recurrent ovarian cancer [96]. Thus, combinational therapeutic strategies have been suggested to inhibit CSCs via blocking the Notch signal pathway; for example, the safety of the combination of RO4929097 and temsirolimus has been proven in solid tumor treatment [97]. The combination of GSIs with trastuzumab and lapatinib could reduce tumor growth and recurrence in breast cancer [98]. It has also been indicated that the combination of GSIs and MAPK could enhance their pesticide effects in CRC cell lines [99].

##### Anti- Delta-like Ligand 4 (DLL4) Monoclonal Antibody

DLL4is known for its function in tumor angiogenesis. The angiogenesis of functional capillaries plays a crucial role in tumor development, and it has been reported that the expression of DLL4 could be regulated by vascular endothelial growth factor (VEGF) on vascular endothelial cells. The increased formation of non-productive angiogenesis could be induced by blocking DLL4; this process showed its anti-tumor potential [100,101]. REGN421 (enoticumab) is a human IgG1 monoclonal antibody, which can inhibit the Notch signal pathway via binding to DLL4; it was observed to inhibit tumor growth by the formation of non-functional capillaries in ovarian tumor xenograft models. The combination of REGN421 with VEGF inhibitor was also suggested as a better strategy in this study [102]. Enoticumab has been proven to be safe and effective in clinical trials [103]. DLL4 inhibitor, either alone or in combination with another DLL4 inhibitor or irinotecan, showed its therapeutic value in xenograft tumor models [104].

#### 3.2.2. Hedgehog(Hh) Signal Pathway Inhibitor

Aberrant activation of the Hh signal pathway is closely related to tumorigenesis, metastasis and recurrence, and it is a good point to improve prognosis by inhibiting the Hh signal pathway. Targets to inhibit the Hh signal pathway include the Hh ligand, Hh receptor and Hh transcription factor, which are an SMO inhibitor, Gli inhibitor and SHH inhibitor, respectively. SMO is an Hh receptor encoded by oncogene smooth; it can mediate Gli to enter the nucleus to initiate the expression of downstream genes. SMO inhibitors have shown their therapeutic value when used alone or in combination with other chemotherapy drugs. Sonidegib has been found to downregulate the expression of SMO and Gli in human natural killer/T-cell lymphoma cells, and researchers have shown its anti-tumor targeting potential [105]. PF-04449913 is an SMO inhibitor, it has been used to treat myeloid malignancies in many clinical trials [106,107]. BMS-833923 is an SMO antagonist, which was identified as a powerful inhibitor in human skeletal stem cells [108]; it was indicated that GFC-0449 could inhibit proliferation and induce apoptosis in CRC cell lines [109]. GFC-0449 was also proved to be a potential treatment in cholangiocarcinoma [110]. Gli is a core transporter factor (TF) in the Hh signal pathway. GANT58 and GANT61 were found as Gli inhibitors, they could inhibit the transcriptional activity of Gli1 in the nucleus, and GANT61 showed better potential in prostatic cell lines [111]. GANT58 was a potential strategy in acute T-cell leukemia cells, and its combined use with AKT inhibitor showed a better therapeutic effect [112]. GANT61 was found to reduce proliferation and cell viability after chemotherapy by the downregulation of CSC-related genes (Oct4, CD44, ALDH and Bmi1) in hepatocellular carcinoma. Zhihua Zhang et al. revealed that GANT61 could induce apoptosis and inhibit proliferation and cell G1/G0 cycle in a dose- and time-dependent manner in multiple myeloma [113], and it was also found to sensitize glioma cells to temozolomide [114]. It was revealed that Arsenic Trioxide could inhibit the growth of human cancer cells by blocking the Hh/Gli pathway in vivo and in vitro [115], and its clinical value has been proven in many studies. There are also some Hh inhibitors whose targets are Hh signal pathway ligands, such as Shh inhibitors, RU-SKI43 and Shh monoclonal antibody 5E1, which have been mentioned in previous studies [116].

#### 3.2.3. Wnt Signal Pathway Inhibitor

It was revealed that the Wnt signal pathway participated in many cancer processes, including glycolysis, glutaminplysis, lipogenesis, the metabolic negative feedback loop and cancer immunotherapy [117], indicating its potential to inhibit tumor development via targeting the Wnt signal pathway. It was summarized that the Wnt signal pathway inhibitors could be divided into two types, secreting inhibitors, which included Dickkopf proteins (Dkks), secreted Frizzled-related proteins (sFRPs), Wnt-inhibitory factor 1 (WIF-1), Wise/SOST, Cgeberus and insulin-like growth factor binding protein 4 (IGFBP-4), and transmembrane inhibitors, which included Shisa, Wnt-activated inhibitory factor 1 (Waif1/5T4), adenomatosis polyposis coli downregulated 1 (APCDD1) and Tiki1 [118]. Many studies have focused on their functions in tumor therapy; Youcheng Shao et al. showed the diagnosis and therapeutic value of Dkks in their study [119]. It was found that sFRPs harbored a biphasic regulating function in the Wnt-βcatenin signal in CSCs [120], the process of which may be determined by the cellular context and concentration of Fzd receptors [121]. Shisa3 was revealed as a tumor suppressor and a new insight for tumor prognosis and therapy [122]; it could be a biomarker for patients suffering from chronic lymphocytic leukemia, regardless of whether they could benefit from lenalidomide treatment [123]. Many Wnt inhibitors have been investigated in many clinical or preclinical trials, such as CGX1321, LGK974 andDKN-01, and some of them have been approved by the FDA to solve clinical problems.

#### 3.2.4. Other Signal Pathway Inhibitors

As mentioned before, CSCs are regulated by many other signal pathways; except for Wnt, Notch and Hh, researchers have also tried to inhibit CSCs via these signal pathways. The TGF-β signal pathway plays a biphasic role in tumor progression; it also plays crucial roles in EMT and TME. Different inhibitors, targeting TGF-β, the TGF-β receptor, the TGF-β ligand and their interaction have been designed in previous studies. The anti-tumor activity of some antibodies, such as 264RAD and GC1008, has been proven; meanwhile, some small molecules that target TβR kinase, such as galunisertib, were also proven to be effective in hepato-cellular cancer and pancreatic cancer [124]; several TGF-β inhibitors have been evaluated in clinical trials. The abnormal activation of the JAK/STAT signal pathway promotes tumorigenesis, and many JAK/STAT signal pathway inhibitors, such as JAK inhibitors (INCB018424 and AZD1480), STAT inhibitors (STAT3 inhibitor and STAT5 inhibitor) and phytochemicals(phenolics, polyphenols, terpenoids, alkaloids, saponins, steroids, lignin and phytoalexin) were found to inhibit tumor growth [125]. PI3K/AKT/mTOR was frequently activated in many human tumors; it also contributes to CSCs in many aspects. Researchers found its feasibility to inhibit tumor development via blocking PI3K/AKT/mTOR, which could be divided into PI3K inhibitor (AZD8835, CUDC907 and GDC0077), AKT inhibitor (ARQ092, AZD5363 and MK2206) and mTOR inhibitor (Ridaforolimus, Sirolimus and Evcrolimus); many of them have been proven to be safe and effective in clinical trials, and some of them, such as Zydelig, Copanlisib, Rapamycin and Sirolimus, have been approved for human cancer treatment by the FDA, except for application in tumor therapy. The off-target effects and some adverse reactions, such as hypoglycemia, pneumonitis and neuropsychiatric effects, are also huge problems to solve in the future [126]. Some PPAR agonists, such as Pioglitazone, also showed tumor inhibition in a previous study [127].

Although the strategies targeting the signal pathway contributed to inhibiting CSCs in many aspects, many related agents have been tested in clinical trials, as shown in Table 3; however, controversial views remain in this field. CSCs are regulated by crosstalk among various signal pathways, and some researchers have found that some agents could promote another tumorgenesis-correlated signal pathway while they working as inhibitors to block the anti-tumor-correlated signal pathway [128]. Toxicity, adverse reactions and drug resistance caused by the gene mutation of components in different signal pathways are also common problems that need to be solved.

### 3.3. Targeting Metabolic Pathways of CSCs

As mentioned before, CSCs harbor a unique energy metabolic pattern. Targeting CSCs metabolism has been found to potentially eliminate CSCs. Metformin and phenformin, which are well known for their anti-diabetes function, were found to inhibit tumor growth via targeting CSCs, including gastric cancer [129], colorectal cancer [130], ovarian cancer [131] and prostate cancer [132]. The mechanism could be explained by its ability to inhibit mitochondrial function. It was revealed that Metformin could reduce tumorigenesis by inhibiting mitochondrial complex I [133]. Cancer therapy benefits from some antibiotics, which have been proven to inhibit the respiratory function of mitochondria; for example, salinomycin showed anti-tumor activity in colorectal cancer [134], and it was also indicated that mitochondrial function could be disrupted by salinomycin [135,136]. ROS are produced in the electron transport chain(ETC) during cell energy metabolism, as found in previous studies, a lower ROS level was observed to maintain CSC properties in CSC niche, thus, eliminating CSCs via a ROS inducer was proposed. Disulfiram and copper could induce apoptosis by increasing the ROS level in ovarian CSCs, which were marked by ALDH^+^ [137]. The manganese(ii)-3,4,7,8-tetramethyl-1,10-phenanthroline complex comprises polymeric nanoparticles and has been indicated to be able to reduce CSC properties by producing ROS in BCSCs [138]. Mitochondrial fatty acid oxidation(FAO) was proven to be necessary in breast cancer stem cells [139]. Etomoxir is an FAO inhibitor; it can suppress tumor development in bladder cancer [140] and nasophanryngeal cancer [141]. Meanwhile, the metabolism of CSCs is influenced by some signal pathway inhibitors; sFRP4 is a Wnt inhibitor, which was revealed to be able to inhibit CSC survival by regulating its metabolism in breast and prostate cancer stem cell lines [142]. Some strategies mentioned in previous studies have focused on transporting drugs to mitochondria efficiently and accurately, for example, the conjugation of targeting mitochondrial drugs to triphenylphosphonium (TPP) or mitochondria-penetrating peptides(MPP) [143]. This strategy also promoted a combination of targeting metabolism drugs with conventional chemotherapy to improve the therapeutic effect. This seems to be a potential strategy to reduce tumors via targeting glycolysis and there have been many studies focused on this field; it was summarized that the main targeting glycolysis therapies include glucose deprivation, GLUT inhibitors, HK-II inhibitors, PFK inhibitors, GAPDH inhibitors, PK-M2 inhibitors and LDH inhibitors [144], and some of them have been proven to have anti-tumor potential. However, this is limited due to its serious adverse systemic reaction in trials, and some of them were even indicated to have a pro-survival role in cancer cells [145]. Thus, the accurate targeting and combination of glycolysis inhibitors must be investigated in more studies and clinical trials. Due to its metabolic heterogeneity in CSCs, some researchers proposed the “two metabolic hit” strategy, during which the glycolysis inhibitors would be used to eliminate CSCs after the first use of mitochondrial inhibitors [146]. Although many metabolic agents have been evaluated in clinical trials (Table 4), an obvious limitation was observed when these agents were used alone. Thus, a combinational strategy consists of different anti-glycolysis and anti-OXPHOS drugs, and conventional chemotherapy drugs might be a better option.

### 3.4. Targeting CSC Niches

The tumor microenvironment (TME) is the shelter where tumor cells live and proliferate, which is composed of matrix components, cellular components and soluble factors. Many studies focused on TME have indicated that TME contributes to proliferation, invasion, metastasis and immune-escaping of tumor cells. In sum, CSCs are tumor cells, CSC niches are also part of TME. There is a complicated interaction between CSC and CSC niches; it was summarized that CAF could regulate the proliferation, expansion and self-renewal via secreting some cytokines, such as CCL2, IGF-1, TGF-β and HMGB-1; what’s more, researchers found that CAF could induce tumor cells dedifferentiated change via secreting HGF. [147] The hypoxia environment caused by the faster proliferation and higher energy demand of CSCs was found to contribute to CSC development, especially in solid tumors [148,149]. Hypoxia-induced factors (HIFs) cause an adaptation for CSCs to become survivable in the hypoxia environment, including changing the metabolic pattern [150] and up-regulating the expression of multipotential genes, such as Oct4 and Sox2 [151]. Many studies have focused on HIFs, especially HIF-α, which was revealed to cause increasing stemness and MDR in colorectal cancer via the GLI 2 signal pathway [152]. The downregulation of HIF-2α could inhibit CSC stemness and induce CSC apoptosis via the AKT-mTOR signal pathway in TNBC [153]; a similar function was also observed in other tumor types. Cancer-associated fibroblasts (CAFs) contribute to the stemness of CSCs by secreting functional proteins and exosomes or activating different CSC-related signal pathways [154]. It was indicated that CAFs could promote stemness properties such as self-renewal, metastasis and chemotherapeutic resistance in CSCs marked by CD24+ [155]. Other components of CSC niches, such as perivascular cells, inflammatory cells, the extracellular matrix (ECM) and some secreted cell factors (VEGF and HGF), were also found to be crucial in promoting proliferation and maintaining stemness in CSCs.

All of the evidence shows the potential to eliminate CSCs by targeting CSC niches. As listed in Figure 3, CXCR4 is a well-known CSC marker in many different cancer types, the crosstalk between CXCR4 and its ligand CXCL12 plays a crucial role in tumor development, also in CSC; it was summarized that CXCL12-CXCR4/CXCR7 axis could maintain CSCs stemness via modulating immune cell migration, recruitment of mesenchymal stem cells, formation of CAFs and vascular endothelial cells, thus, it’s a potential strategy to inhibit CSC niches via targeting CXCL12-CXCR4/CXCR7 axis [156]; some CXCR4/CXCR7 inhibitors listed in Table 5 have been evaluated in different clinical trials. Meanwhile, it may be another way to disrupt CSC niches via inhibiting HIF, a feedback loop composed of HIF-1α and SENP1 existed in the hepatocellular carcinoma, and it was indicated that the positive feedback loop was closely related to the increasing stemness of hepatocellular cancer stem cells and a potential target for HCC therapy [157]. The nanomaterial, called Gd@C82(OH)22, was found to enhance CSC elimination by inhibiting HIF-1αand TGF-βin breast cancer stem cells [158]. Many HIF inhibitors have been tested their antitumor effect in preclinical and clinical trials, although there is no novel evidence to indicate that these HIF inhibitors work through inhibit CSC niche, it may be a potential mechanism which needs to be found in the future, we list some HIF inhibitors in Table 5. Targeting CAFs is another effective strategy to target CSC niches; there is a crosstalk between CAFs and CSCs, and many strategies have been designed from the crosstalk, including inducing the depletion of CAFs harboring CSC-supporting activities and targeting the signal pathway existing in the CAFs–CSCs crosstalk [154]. Many CAFs–CSCs crosstalk-related signal pathway inhibitors have shown their inhibiting function in CAFs, such as LGK974 and OPB-31121. CAFs could be marked by some cell surface molecules, and it has been revealed that CAFs marked by CD10+ or GPR77+ could be ideal targets to inhibit CSCs in solid tumors [159]. As mentioned before, angiogenesis contributes to CSCs survival; it is a good strategy to eliminate CSCs via inhibiting angiogenesis in CSC niches. Vascular endothelial growth factor (VEGF) plays a crucial role in tumor development, and many studies have shown its therapeutic function in CSCs via blocking VEGR or VEGFR [160]. Endothelial mesenchymal transformation (EMT) is a key characteristic of stem cells; it plays a crucial role in carcinogenesis. Tumor cells show stronger capabilities of proliferation, differentiation and invasion via regulating EMT, and the EMT phenotype contributes to the MDR of tumor cells; it is also closely related to CSCs in many aspects. Researchers have revealed that the expression of surface markers of CSCs, such as CD44 and ALDH, could be regulated by the expression of transcription factors of EMT, such as TWIST, SNAIL and SLUG [161,162,163]. EMT also shares many of the same signal pathways, such as TGF-β and Hh with CSCs and it was proposed that EMT may be the reason for the acquisition or maintenance of the stemness of CSCs; it was found that the stemness of CSCs and EMT could be regulated by some signal pathway inhibitors, and targeting the EMT phenotype could potentially inhibit tumor cells, even CSCs. GSK3β inhibitors have been identified as EMT inhibitors; it was indicated that GSK3β inhibitors acted as selective inhibitors of EMT and CSCs in TNBC [164]. Some micro-RNAs have proven their function in EMT and CSC regulation in previous studies [165], including some onco-miRNA, such as miR-191, in promoting EMT and increasing the stemness of CSCs in breast cancer [166], and some anti-onco-miRNA, such as miR-34a, in inhibiting EMT and CSCs in head and neck squamous cell carcinoma (HNSCC) [167]. Some other strategies have also been proposed, such as targeting chemokine receptors to inhibit CSCs in CSC niches. Obviously, targeting CSC niches is a potential strategy to inhibit CSCs; some agents, such as VEGF inhibitors and HIF inhibitors have been proven to be valid and safe in many clinical trials (Table 5), CSC niches form a complicated environment for CSCs; thus, more detailed components in the niche and more specific targets need to be found to target CSCs in further studies and clinical trials.

### 3.5. Targeting Differentiation Mechanisms of CSCs

CSCs harbor the properties of unlimited self-renewing and multiple directions differentiation, these dedifferentiated properties contribute to the development, recurrence after treatment and MDR of cancer to a large degree. The higher dedifferentiated states of cancer cells were always linked to the poorer prognosis, a contrary result to the differentiated states. As with iPSCs, the source of CSCs could be some stromal cells induced by some reprogramming factors; it was indicated that de-differential states of CRC CSCs derived from stromal fibroblasts could promote chemoresistance in CRC [168]. Thus, inducing cell differentiation from a dedifferentiated state could be a feasible therapy [169] (Figure 4). Inducing differentiation therapy has been proven to be successful in many diseases, including orthopedic treatment, age-related bone loss and some hematological tumors. Many related agents and biological technologies have been proven to be effective in cellular and clinical trials. Sandy Azzi et al. showed that IL15 could induce renal cancer stem cell differentiation and increase sensitivity to chemotherapeutic drugs [170]. Low-intensity ultrasound can modulate cell proliferation and induce cell differentiation; it was potentially shown that the stemness of liver cancer stem cells marked with CD44/CD133 could be impaired by dual-frequency ultrasound with its ability to induce CSC differentiation [171]. Nanomedicine shows faster, more accurate and sensitive characteristics in medicine. A combination of nanoparticle engineering and hypothesis-free sensing was proposed to induce CSC differentiation and showed its potential functions in antitumor therapeutics [172]. All-trans retinoic acid (ATRA) is the mesostate of vitamin A, which was known as its promoting differentiation in many disease treatments, especially in malignant tumors [173]. There are also many other differentiation inducers, such as dimethyl sulfoxide and retinal, although there is no clear evidence to support their relationship with CSC, their application has been proven to be valuable in antitumor therapy, inducing CSCs differentiation maybe the potential mechanism, and some of them have been evaluated in many clinical trials (Table 5).

### 3.6. Immunity Method of CSCs

Tumor immunological therapy has shown a large amount of potential in many cancer types. The mechanism of tumor immunological therapy is to inhibit or eliminate tumor cells by activating the immune system, which includes innate immunity and adaptive immunity. The recent evidence shows that CSCs harbor the capacity of immuno-suppression via several mechanisms; it was revealed that some well-known immunosuppressive molecules, such as PD-L1 and CTLA-4, were higher expressed in CSCs than in common tumor cells [174,175]. Antigen presentation plays a crucial role in cells immunity, it was indicated that CSCs could impair the antigen presentation by downregulating the expression of MHC molecules and some antigen processing molecules, such as transporter associated with antigen processing (TAP) and β-macroglobulin [176]; researchers also found that the abnormal expression of tumor-associated antigens in CSCs contributed to immune invasion via reducing immunogenicity; CSCs could also promote to build suppressive TME via recruiting some immune cells, such as tumor-associated macrophage (TAM) and regulatory T cell(Treg), and secreting some immuno-suppressive molecules, such as CD200, TGF-βand IFNs, to inhibit CSCs-targeted immune-response [177], Hiroyuki Tsuchiya et al. also summarized that CSCs could impair NK cell function, such as inhibiting the release of cytolytic granules, regulating the expression of the different NK-cell ligands, to escape immune response, they thought the characteristic of immune invasion was a more fundamental feature to CSCs than the tumor-initiating property [178]. It is necessary to find some new strategies to target CSCs via immunity methods. As mentioned before, except for the surface markers, such as CD44, CD133 and ALDH, which could be targeted by some immunological molecules. There are also some other immunological strategies to target CSCs.

#### 3.6.1. Tumor Vaccine

The tumor vaccine has shown a large amount of potential and success in recent years; its basic mechanism is to inhibit or eliminate tumor cells by activating the human immune system, which is initiated by the artificial injection of immunogenic tumor antigen. With the increased and deeper research into the tumor vaccine and CSCs, a CSC-based vaccine was proposed to target and eliminate CSCs directly. Lin Lu et al. indicated that CSCs marked by ALDH could be directly targeted by the ALDH CSC-DC vaccine, and also showed its potential function in the adjuvant setting of recurrent tumors [179]. MUC1 was found to participate in regulating the stemness of colorectal stem cells (CCSCs), and the CSC vaccine based on MUC1 has been proven to be effective and safe in inhibiting CCSCs marked by CD133+ via activating humoral immunity [180]. A similar CSC inhibiting effect of the CSC vaccine was also found in melanoma, ovarian cancer, lung cancer, liver cancer and nasopharyngeal cancer. CSCs comprise a small percentage of the total tumor tissue, which makes it more difficult to target CSCs by a tumor vaccine. Some combinational strategies have been suggested to target CSCs more efficiently, such as a combination of the CSC vaccine with some conventional chemotherapeutic drugs [181]. In recent years, some CSC vaccines have been evaluated in clinical trials; however, many studies focusing on CSC vaccines were conducted in immunodeficient mice where the immunological environment was significantly different from the normal human body. Thus, more clinical trials are necessary to prove its successful application.

#### 3.6.2. Chimeric Antigen Receptor T Cell Therapy (CAR-T)

CAR-T has become popular in target therapy in recent years; T cells can be reprogrammed into CAR-T cells by adding an artificially designed CAR with gene-editing technology, which makes T cells more accurate and threatening to target tumor cells. To date, there have been five generations of CAR designing. CAR-T first showed its success in the hematological tumor [182]; it has also been found to be effective in many solid tumors, including gastric cancer, pancreatic cancer and breast cancer. Researchers thought it was feasible to inhibit CSCs by CAR-T, some surface markers, such as CD44, CD133 and EpCAM, were used as targets to identify the function of CAR-T in CSC therapy. The efficiency and safety of some CAR-T products have been evaluated in clinical trials. Although CAR-T harbors many advantages in antitumor therapy, there are still many issues, such as the lack of a unique specific target, effective concentration and persistence of CAR-T cells in the targeting area; meanwhile, some adverse reactions, such as off-tumor effects, neurotoxicity, cytokine release syndrome(CRS), soluble tumor syndrome and blood coagulation disorders, which have been mentioned in previous studies [183,184]. The majority of CSC–CAR-T studies are preclinical trials, cell experiments and animal experiments; however, more related clinical studies are needed to prove its value via working alone or in combination with tumor target therapy.

#### 3.6.3. Oncolytic Virus

The oncolytic virus was designed to damage tumor cells by viruses with low toxicity. Its basic mechanism includes inducing cell lysis via virus proliferation and toxicity of virus-related proteins, activating non-specific and specific immunity and inhibiting angiogenesis. The first oncolytic virus-related report was the rabies virus, which could inhibit cervical cancer cells. Subsequently, many studies indicated that oncolytic virus therapy showed potential in the anti-tumor field [185,186], including hematological malignancies and some solid tumors. The oncolytic virus was also proven to be efficient in tumor therapy via inhibiting CSCs. As mentioned before, conventional chemotherapeutic drugs showed failure in eliminating CSCs due to several factors, such as quiescent state, anti-apoptotic proteins, ROS level and DNA repairmen. However, the oncolytic virus could induce CSC lysis via infecting both quiescent and dividing cells, this process will not be influenced by these factors [187]. Although the oncolytic virus showed a large amount of potential in inhibiting CSCs, a combined therapy of oncolytic virus with chemotherapy and radiotherapy should be more feasible and efficient in CSC therapy, which has been conducted and proven to be efficient in many cases. In addition, the sensitivity and susceptibility of oncolytic virus to host tumor cells are still core problems in engineering oncolytic virus.

#### 3.6.4. Targeting CSC-Immune Cell Crosstalk

As we know, there is a close link between CSC biological characteristics and the immune system. In addition to differentiating into endothelial cells, pericytes and fibroblasts to build CSC niches, CSCs can also build their own TME via regulating various different immune cells and signal pathways, which contributes to tumor immunosuppression and immune escape. According to a recent report, CSCs could recruit tumor-associated macrophages (TAMs) and induce their polarization via different chemokines (IL-4 and IL-13) and signal pathways (Wnt, STAT3 and NF-kB). CSCs can promote bone marrow-derived macrophage (BMDM) invasion and activation via secreting various soluble cytokines and exosomes, and can also inhibit T-cell functions via secreting TGF-β, CCL2, TNC, etc. Meanwhile, these immune cells can also regulate CSC stemness via different cytokines. Peiwen Chen et al. highlighted the therapeutic potential of targeting CSC–TAM crosstalk, CSC–MDSC crosstalk and CSC–T-cell crosstalk in their study [188]. Karina E Gomez et al. found that TAM could increase hyaluronic acid (HA) and tumor cell invasion, whereas HA could enhance PI3K-4EBP1-SOX2 signaling and CSC fraction in human neck squamous cell carcinoma (HNSCC); they found that it is feasible to inhibit CSCs via targeting CD44 or/and TAM [189]. MDSC contributes to immune suppression. Dongjun Peng et al. revealed that MDSC could enhance CSC stemness and inhibit T-cells activation via the STAT3-NOTCH signal pathway in breast cancer; they found its potential to inhibit CSCs and immune escape via targeting STAT3-NOTCH crosstalk [190]. It was indicated that MDSC could increase CSC stemness and tumor cell PD-L1 expression by the production of PGE2 in epithelial ovarian cancer, and also MDSC might be an effective target to inhibit ovarian tumor cells via reducing ovarian CSC stemness and PD-L1expression [191]; meanwhile, Xiaofeng Li et al. revealed that MDSC could also promote CSC stemness in ovarian cancer via inducing the CSF2/p-STAT3 signaling pathway, and they thought it could enhance the efficacy of conventional treatments by targeting MDSC and colony-stimulating factor 2 (CSF2) [192]. These preclinical studies indicate the therapeutic potential of targeting CSC–immune cells crosstalk in CSC therapy.

To date, many immunity products targeting CSCs have been evaluated in clinical trials (Table 6). There are also some other existing immunity-related methods targeting CSCs, such as immune checkpoint inhibitors (PD-1/PD-L1 inhibitors), immune activators and CAR-NK, which have been mentioned in previous studies. Immunity is the basic mechanism to protect the human body from damage, immunotherapy should be the basic strategy to inhibit CSCs alone or combined.

## 4. Conclusions

Obviously, CSCs play a crucial role in the tumorigenesis, development, metastasis and recurrence of tumors. The adaptive mechanism consists of a unique mitosis pattern, unique metabolic phenotype, and CSC niches, which protects CSCs and makes them difficult to eliminate by conventional therapies. According to previous studies, many problems still need to be solved: (1) The lack of strong specificity to target CSCs: CSC comprises a small percentage in total cancer tissues; meanwhile, some of the surface markers detected on CSCs have also been found on normal stem cells, even normal cells, which makes accurate surface maker-targeted therapies more difficult and causes adverse damage to the human body. (2) As with TME in solid tumor cells, CSC niches are a stubborn shield to protect CSCs from human immunity attack, more detailed components of CSC niches and their targeted strategies need to be found to impair CSC niches. (3) CSCs are also tumor cells, many tumor-related signal pathways must contribute to CSCs in some aspects. CSC-related signal pathways work in a complex network, and the unclear synergism or antagonism among different signal pathways makes signal pathway-targeting strategies controversial. The signal pathway is just the route, not the origin; thus, more attention should be paid to find a strategy which could inhibit CSCs at their origin. (4) Most of the products proven their anti-CSCs in vitro experiment where the immunity environment was too simple; therefore, more clinical trials need to be conducted to test the safety and efficacy of anti-CSC–products. Immunotherapy should be considered as the baseline. In various immunotherapies, such as CAR-T, CAR-NK and DC-vaccine, they have shown anti-tumor potential in recent years. Additionally, nanomedicine provides an accurate route to transport drugs to the targeted areas; a combinational strategy consisting of different drugs that work with different mechanisms may be an ideal option in inhibiting CSCs.

## Figures and Tables

**Figure 1 cancers-13-04814-f001:**
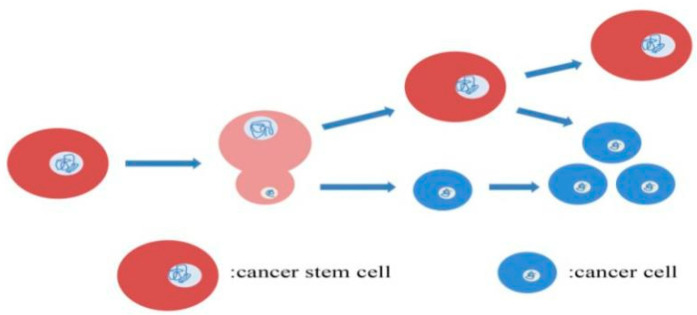
Asymmetric mitotic division pattern of CSCs.

**Figure 2 cancers-13-04814-f002:**
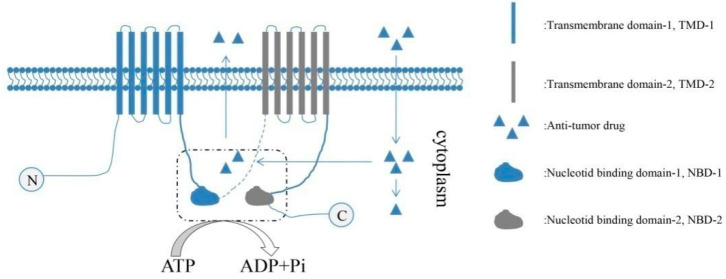
An ABC transporter is composed of two transmembrane domains (TMD-1 and TMD-2) and two nucleotide-binding domains (NBD-1 and NBD-2). As shown in the figure, chemotherapeutic drugs could be pumped out of tumor cells via the ABC transporter expressed on CSCs.

**Figure 3 cancers-13-04814-f003:**
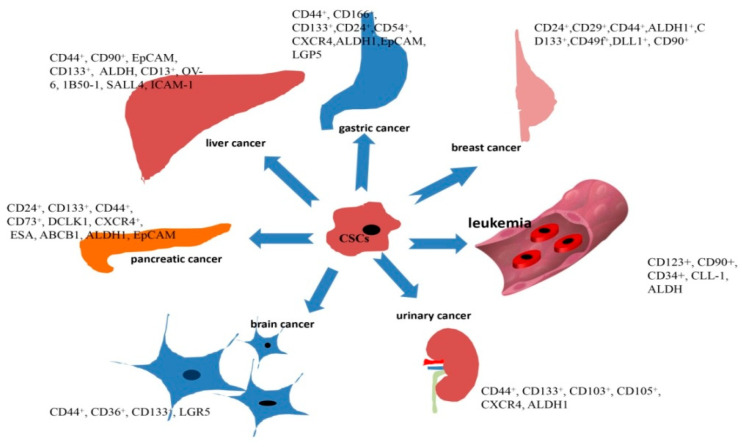
Surface markers expressed on different CSCs.

**Figure 4 cancers-13-04814-f004:**
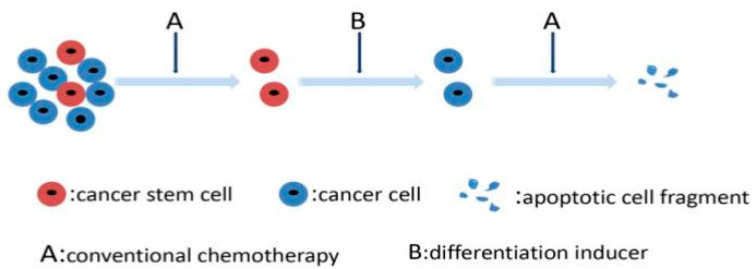
Inducing differentiation therapy to eliminate CSCs.

**Table 1 cancers-13-04814-t001:** Some agents targeting surface markers of CSCs which have been evaluated in clinical trials.

Agents	Target	NCT Number	Phase	Current State	Condition
BIWA1	CD44v6	NCT02204046	I	Completed	Breast cancer
		NCT02204059	I	Completed	Lung cancer
RO5429083	CD44	NCT01358903	I	Completed	Malignant solid tumors
		NCT01641250	I	Completed	Leukemia
17-1A	EpCAM	NCT02915445		Recruiting	Nasopharyngeal/breast
					cancer
		NCT00309517		Terminated	Rectal carcinoma
Catumaxomab	EpCAM	NCT01815528	II	Completed	Ovarian cancer
		NCT01246440	II	Completed	Ovarian cancer
		NCT00326885	II	Completed	Malignant ascites
		NCT00464893	II	Completed	Gastric cancer
Adecatumumab	EpCAM	NCT00866944	II	Completed	Colorectal liver
					Metastases
ING-1	EpCAM	NCT00051675	I	Completed	Adenocarcinomas
Edrecolomab	EpCAM	NCT00002968	III	Completed	Colon cancer
CSL-360	CD123	NCT00401739	I	Completed	Acute Leukemia
Talacotuzumab	CD123	NCT02472145	II/III	Completed	Acute Leukemia
anti-ICAM-1 Mab	CD54	NCT01025206	I	Completed	Multiple myeloma

**Table 2 cancers-13-04814-t002:** Surface markers expressed on the CSCs could also be expressed on the normal organs/cells.

Surface Marker	CSCs	Normal Cells/Organs
CD44	Liver [42], stomach [43], breast [44], pancreas [45], glioma [46], kidney [47], colon, rectum [48]	Lymphocytes [62], vascular endothelial cells [63] corneal cells [64], astrocytes [65], mucosal cells [66] intestinal epithelial cells [67]
CD29	squamous cell carcinoma [49], breast [50], colon [51]	vascular endothelial cells [68], salivary glands [69] myoepithelial cells [70], epithelial cells [71]
CD54	prostate [52], breast [53]	hematopoietic cells [72,73]
CD123	leukemic stem cells [54]	testis [74], lung, brain [75], basophil [76]
ALDH	stomach [55], breast [56], pancreas [57], kidney [58], colon [59] squamous cancer [60]	breast stem cells [77], renal tubular stem cells [78]
CD73	pancreas [61]	lymphatic endothelial cells [79], vascular endothelial cells [80]

**Table 3 cancers-13-04814-t003:** Some agents targeting CSCs-related signal pathways which were evaluated in clinical trials.

Drug Name	Target	NCT Number	Phase	Current State	Condition
RO4929097	Notch	NCT01120275	II	Terminated	Melanoma
		NCT01238133	I	Terminated	Breast cancer
		NCT01175343	II	Completed	Ovarian cancer
		NCT01154452	I/II	Completed	Sarcoma
		NCT01198535	I	Terminated	Colorectal cancer
		NCT01232829	II	Completed	Pancreatic cancer
		NCT01122901	II	Terminated	Glioblastoma
		NCT01131234	I	Completed	Solid tumors
LY900009	Notch	NCT01158404	I	Completed	Advanced tumors
MK-0752	Notch	NCT00645333	I/II	Completed	Breast cancer
		NCT00106145	I	Completed	Breast cancer
		NCT00572182	I	Terminated	CNS cancer
OMP-54F28	Wnt	NCT02069145	I	Completed	Hepatocellular cancer
		NCT02092363	I	Completed	Ovarian cancer
		NCT02050178	I	Completed	Pancreatic cancer
		NCT01608867	I	Completed	Solid tumors
LGK974	Wnt	NCT01351103	I	Recruiting	Malignancies
WNT974	Wnt	NCT02278133	II	Completed	colorectal cancer
DKN01	Wnt	NCT04681248		Available	solid tumors
		NCT01457417	I	Completed	Lung cancer
		NCT03645980	I/II	Recruiting	liver cancer
BMS-833923	Hh	NCT00670189	I	Completed	Advanced cancer
		NCT01413906	I	Completed	Solid tumors
		NCT01357655	II	Completed	Chronic myeloid leukemia
GDC-0449	Hh	NCT01088815	II	Completed	Pancreatic cancer
		NCT00607724	I	Completed	solid tumors
AVID200	TGF-β	NCT03834662	I	Active	Malignancies
LY2157299	TGF-β	NCT02452008	II	Recruiting	prostatic cancer
		NCT02240433	I	Completed	Hepatocellular cancer
		NCT02688712	II	Recruiting	Rectal cancer
GC1008	TGF-β	NCT01112293	II	Completed	Mesothelioma
		NCT01401062	II	Completed	breast cancer
Itacinib	JAK-STAT	NCT04358185	II	Recruiting	hepatocellular cancer
		NCT03989466	I	Recruiting	T-Cell prolymphocytic
					Leukemia
Ruxolitinib	JAK-STAT	NCT04906746	I	Not recruiting	Lung cancer
		NCT01895842	I	Completed	Leukemia
		NCT03514069	I	Recruiting	Glioma
		NCT01594216	III	Completed	breast cancer
SB1518	JAK-STAT	NCT00719836	I/II	Completed	Myeloid malignancies
		NCT04635059		Recruiting	Prostate cancer
		NCT02323607	I	Completed	Acute myeloid leukemia
Alpelisib	PI3K/AKT/mTOR	NCT04526470	II	Not recruiting	Gastric cancer
		NCT04544189	II	Recruiting	Breast cancer
		NCT01300962	I	Completed	Breast cancer
Temsirolimus	PI3K/AKT/mTOR	NCT01072890	I	Completed	Solid tumors
		NCT01050985	I	Completed	Advanced malignancies
Copanlisib	PI3K/AKT/mTOR	NCT03498430	I	Completed	Non-Hodgkin’s lymphoma
		NCT04750941	II	Recruiting	Endometrial cancer
Capivasertib	PI3K/AKT/mTOR	NCT04742036	I	Recruiting	Solid Tumors
		NCT04087174	I	Completed	Prostate cancer
		NCT04862663	I/III	Recruiting	Breast cancer
Pioglitazone	PPAR	NCT02133625	I	Completed	Solid tumors

**Table 4 cancers-13-04814-t004:** Some agents targeting metabolic pathways of CSCs which have been evaluated in clinical trials.

Agents	Target	NCT Number	Phase	Current State	Condition
		NCT00897884		Completed	Breast cancer
		NCT01266486	II	Completed	Breast cancer
Metformin	OXPHS	NCT01243385	II	Completed	Prostate cancer
		NCT02437656	II	Completed	Rectal cancer
		NCT03359681		Recruiting	Colon cancer
Disulfiram	OXPHS	NCT01118741		Completed	Prostate cancer
		NCT02678975		Completed	Glioblastoma
		NCT03584009	II	Completed	Breast cancer
		NCT03000257	I	Not yet recruiting	Solid tumors
Venetoclax	BCL-2	NCT03082209	I	Recruiting	Solid tumors,
					Hematologic malignancies
		NCT04161885	III	Recruiting	Acute myeloid leukemia
		NCT02265731	I/I	Completed	Hematological malignancie
Ketoconazole	HK-II	NCT03763396		Not yet recruiting	Glioma
		NCT01036594		Completed	Prostate cancer
2-DG	Glut	NCT00096707	I	Completed	Solid tumors

**Table 5 cancers-13-04814-t005:** Some agents targeting CSC niches and inducing differentiation which have been evaluated in clinical trials.

Agents	NCT Number	Phase	Current State	Condition
**CSCs niche inhibitor**				
LY2510924	NCT02652871	I	Completed	Acute myeloid leukemia
	NCT01439568	II	Completed	Lung cancer
BKT140	NCT01010880	I/II	Completed	Multiple myeloma
AMD3100	NCT00512252	I/II	Completed	Acute myeloid leukemia
BL-8040	NCT01838395	II	Completed	Acute myeloid leukemia
Bevacizumab	NCT01190345	II	Completed	Breast cancer
	NCT01137968	II	Completed	Lung cancer
	NCT03632798	III	Recruiting	Ovarian cancer
Topotecan	NCT00320983	I	Completed	Cervical cancer
	NCT00477282	III	Completed	Epithelial ovarian cancer
	NCT01630018	II	Completed	Ovarian cancer
Digoxin	NCT01162135	II	Completed	Prostate cancer
	NCT00650910	I	Completed	Breast cancer
	NCT02106845	I	Completed	Solid tumors
PT2385	NCT03216499	II	Completed	Recurrent glioblastoma
	NCT04989959	I	Recruiting	Renal cell cancer
EZN-2986	NCT01120288	I	Completed	Solid tumors
	NCT00466583	I	Completed	Solid tumors/lymphoma
**Differentiation inducer**				
retinoic acid	NCT01276730	II	Completed	Cervical cancer
	NCT00002586	II	Completed	Lung cancer
	NCT01048645	II	Completed	Lung cancer
	NCT00004149	II	Completed	Prostate cancer
arsenic trioxide	NCT00128596	II	Completed	Metastatic liver cancer
	NCT00005069	II	Completed	Metastatic kidney cancer
dimethylsulfoxide	NCT04439318	II	Not yet recruiting	Multiple myeloma

**Table 6 cancers-13-04814-t006:** Immunity methods targeting CSCs which have been evaluated in clinical trials.

Agents	Target	NCT Number	Phase	Current State	Condition
	CD133	NCT02541370	I/II	Completed	Advanced malignancies
**CAR-T**	CD44v6	NCT04097301	II	Recruiting	AML, MM
		NCT04427449	I/II	Recruiting	Tumors (CD44v6+)
	EpCAM	NCT03563326	I	Recruiting	Gastric cancer
		NCT02915445	I	Recruiting	Nasopharyngeal cancer,
					Breast cancer
		NCT04151186		Not yet recruiting	Recurrent/Refractory
					Solid tumors
		NCT02729493	II	Unknown	Liver cancer
	CD19	NCT04532281	I	Recruiting	AML, Non-Hodgkin’s
					lymphoma
		NCT02975687	I	Completed	AML
		NCT04833504		Completed	B Cell lymphoma
		NCT03811457		Completed	Leukemia, lymphoma
	CD123	NCT04272125	I/II	Recruiting	AML
		NCT04265963	I/II	Recruiting	AML
	CD34	NCT03473457		Recruiting	Relapsed/Refractory AML
	CXCR4	NCT04727008	I	Not yet recruiting	Refractory/Relapsed MM
**DC vaccine**	ALDH	NCT02176746	I/II	Completed	Colorectal cancer
		NCT02178670	I/II	Completed	Ovarian cancer
		NCT02063893		Completed	Breast cancer
		NCT02084823	I/II	Completed	Lung cancer
		NCT02115958	I/II	Completed	nasopharyngeal cancer
		NCT02074046	I/II	Completed	pancreatic cancer
		NCT02089919	I/II	Completed	liver cancer
	Unknown	NCT00846456	I/II	Completed	Glioblastoma

Acute myeloid leukemia (AML), Multiple myeloma (MM).

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
