# Peer review of "Cancer Stem Cell for Tumor Therapy"

_cancers, 2021, doi:10.3390/cancers13194814_

Round 1
Reviewer 1 Report
I am satisfied with all changes done to the manuscript.
Author Response
Point: I am satisfied with all changes done to the manuscript.
Response: We gratefully thank you for your positive and constructive comments and suggestions on our manuscript, and we`ll keep learning to improve our work in the future.
Reviewer 2 Report
Dear authors, I appreciate your efforts to implement the manuscript, describing some more recent data on relevant topics related to CSCs in tumour, for therapeutic purposes. The paper gives a more comprehensive picture on the matter. Nevertheless, it still requires an extensive editing of the language, since many concepts could be more clearly explained. For instance, you should be more uniform in the titles: if you write 'Targeting signal pathways [of CSCs maybe]' in the paragraph 3.2, thus you should write 'Targeting differentiation mechanisms of CSCs' instead of 'Inducing differentiation method'. Also, in the tables you frequently write: 'Some agents targeting CSC ..... which have been evaluated in clinical trials'. Are they the most recent agents tested for therapy? You can also specify the link to your search in clinicaltrials.gov (e.g.). further, you should insert references to support your sentences (e.g. ‘Some combinational strategies have been suggested to target CSCs more efficiently, such as a combination of the CSC-vaccine with some conventional chemotherapeutic drugs or radiotherapy’ – lines 545-547, just as an example). More recent literature has not been mentioned in your work, and maybe it could be appropriate [e.g. https://doi.org/10.3389/fonc.2019.01104, or 10.3390/cancers11050732].
Author Response
Point 1: It still requires an extensive editing of the language, since many concepts could be more clearly explained. For instance, you should be more uniform in the titles: if you write 'Targeting signal pathways [of CSCs maybe]' in the paragraph 3.2, thus you should write 'Targeting differentiation mechanisms of CSCs' instead of 'Inducing differentiation method'.
Response 1: we appreciate your valuable comment on our manuscript, “Targeting differentiation mechanisms of CSCs” really is a better title, in addition, to make titles in our manuscript more uniform, we have changed the titles, such as '3.1 Targeting surface markers of CSCs', '3.2 Targeting signal pathways of CSCs' and '3.6 immune method of CSCs'. Meanwhile, to meet the language standard of journal, we have checked our manuscript again and we also have chosen MDPI`s English editing service to improve our manuscript, you can find the English editing certificate in attachment.
Point 2: In the tables you frequently write: 'Some agents targeting CSC ..... which have been evaluated in clinical trials'. Are they the most recent agents tested for therapy? You can also specify the link to your search in clinicaltrials.gov (e.g.).
Response 2: In fact, the agents mentioned in the tables are those which have been tested their safety and efficiency in clinical trials(maybe not the most recent agents). In order to provide comprehensive information for readers, different NCT numbers are detailed listed in every Table, readers can find every clinical trials via searching NCT number in clinicaltrials.gov.
Point 3: You should insert references to support your sentences (e.g. ‘Some combinational strategies have been suggested to target CSCs more efficiently, such as a combination of the CSC-vaccine with some conventional chemotherapeutic drugs or radiotherapy’ – lines 545-547, just as an example).
Response 3: Thanks for your valuable comment on our manuscript, which can make our manuscript more convincing and rigorous. We have added some recent references to support our sentences in the manuscript, such as Ref. 147/150/151/156/181/183/184.
Point 4: More recent literature has not been mentioned in your work, and maybe it could be appropriate .
Response 4: Thanks for your comment on our manuscript, those papers you mentioned really can improve our manuscript, especially the interaction between CSC and its niche components(CAF and hypoxia). Meanwhile, we have learned some more recent literatures about CSC niches, we describe the potential of targeting CXCL12-CXCR4/CXCR7 axis in CSC niches, and we also add some new sentences and date to make our manuscript more useful for readers.
Thank you again for your constructive comments and suggestions on our manuscript, and we look forward to receiving your further comments.

Reviewer 3 Report
All the comments have been taken into account in improving the quality of the article.
Author Response
point: All the comments have been taken into account in improving the quality of the article.
Response: We gratefully appreciate your valuable comments on our manuscript, and we`ll keep learning to improve our work in the future.
This manuscript is a resubmission of an earlier submission. The following is a list of the peer review reports and author responses from that submission.
Round 1
Reviewer 1 Report
In the manuscript # cancers-1280169, by Huang and colleagues, “Cancer Stem Cell for Tumor Therapy”, authors aim to provide an overview of the biological characteristics of CSCs, including mitotic pattern, metabolic phenotype, therapeutic resistance and related mechanism. Authors count the recent methods to identify and isolate CSCs from tumor tissue for scientific research. Furthermore, they summarize the CSCs targeted strategies, including targeting CSCs surface markers, targeting CSCs related signal pathways, targeting CSCs niche, targeting metabolic pathways, inducing differentiation therapy and immunotherapy, including immunity checkpoint inhibitor, tumor vaccine, CAR-T, and Oncolytic virus. Authors highlight the potential of immuno-therapy and its combinational anti-CSCs therapies, which are composed of different drugs working in different mechanisms for tumor treatment. This is a potentially interesting manuscript that requires substantial improvements.
Improvements suggestions:
1.)
Besides table 1, authors should provide well-referenced tables for the remaining paragraphs describing treatment options.
2.)
The English, although mostly grammatically correct, sounds quite artificial. I would recommend joining efforts with a native speaker-author from Europe, North America, or Australia that would be willing to polish manuscript’s language.
Reviewer 2 Report
The review aims to discuss the current knowledge on therapeutic strategies targeting Cancer Stem Cells, presenting both promising results and limitations for clinical application.
The argument is not coherently exposed, and is not comprehensive, the manuscript is not clearly written.
The paragraphs should be completed with more recent literature, and should briefly describe all data, at least on the major topics [i.e. not all the major signaling pathway activated in CSC have been described, and EMT has just been mentioned, without discussing at all the involvement of this process in CSC and the implications for tumour therapy], in order to give a scientific significant contribution in the field. The arguments should be more clearly presented and in a more orderly pattern.
I suggest authors to substantially revise the manuscript to be published in Cancers.
Reviewer 3 Report
The present article aims at reviewing some progress in cancer stem cells general understanding and on current therapeutic approach aiming to target this resilient population. Although some topics are well addressed producing a coherent argument about the subject and a focused description of the field, major improvements must be provided in other sections listed below. Furthermore, an extensive editing must be performed for English soundness.
- In the paragraph: “Mitotic division pattern of CSC”, the authors must provide a more detailed description about the mechanism correlated with CSC mitosis, such as niche role on CSC fate determination (stem vs differentiated daughter), mitotic spindle assembly.
- The topic, “ Targeting cell surface molecules – Monoclonal antibody”, needs a more insightful description by the authors. In particular, they must provide a more specific background information, a more detailed relevance in the field, address how those results shape the current understanding of the topic and lay the ground for possible future interventions.
- The topic “Immunity method” needs a more exhaustive introduction describing the immunological characteristics of CSC correlated with immunosuppression and immune resistance.